

# Identification and molecular characterization of the alternative spliced variants of beta carbonic anhydrase 1 (βCA1) from *Arabidopsis thaliana*

Jinyu Shen[1,*], Zhiyong Li[2,3,*], Yajuan Fu[1] and Jiansheng Liang[1,3]

[1] Department of Biology, Southern University of Science and Technology, Shenzhen, China
[2] Academy for Advanced Interdisciplinary Studies, Southern University of Science and Technology, Shenzhen, China
[3] Key Laboratory of Molecular Design for Plant Cell Factory of Guangdong Higher Education Institutes, Department of Biology, Southern University of Science and Technology, Shenzhen, China
* These authors contributed equally to this work.

## ABSTRACT

Carbonic anhydrases (CAs) are ubiquitous zinc metalloenzymes that catalyze the interconversion of carbon dioxide and bicarbonate. Higher plants mainly contain the three evolutionarily distinct CA families αCA, βCA, and γCA, with each represented by multiple isoforms. Alternative splicing (AS) of the CA transcripts is common. However, there is little information on the spliced variants of individual CA isoforms. In this study, we focused on the characterization of spliced variants of βCA1 from *Arabidopsis*. The expression patterns and subcellular localization of the individual spliced variants of βCA1 were examined. The results showed that the spliced variants of βCA1 possessed different subcellular and tissue distributions and responded differently to environmental stimuli. Additionally, we addressed the physiological role of βCA1 in heat stress response and its protein-protein interaction (PPI) network. Our results showed that *βCA1* was regulated by heat stresses, and *βca1* mutant was hypersensitive to heat stress, indicating a role for βCA1 in heat stress response. Furthermore, PPI network analysis revealed that βCA1 interacts with multiple proteins involved in several processes, including photosynthesis, metabolism, and the stress response, and these will provide new avenues for future investigations of βCA1.

Corresponding authors
Zhiyong Li, lizy6@sustech.edu.cn
Jiansheng Liang,
liangjs@sustech.edu.cn

## INTRODUCTION

Carbonic anhydrases (CAs) are a group of Zn-containing enzymes that catalyze the reversible hydration of carbon dioxide ($CO_2$), generating proton ($H^+$) and bicarbonate ($HCO_3^-$). As $CO_2$ is the main source of carbon, these CA enzymes are involved in crucial physiological processes, including almost all metabolic processes in higher plants and algae. Molecular, biochemical, and genetic studies of CAs in a wide range of tissues across

diverse plant species have determined that CAs are involved in a wide range of diverse biological processes, including the provision of bicarbonate for anaplerotic reactions (*Werdan & Heldt, 1972*; *Espie & Kimber, 2011*), gas and ion exchange (*Jacobson, Fong & Heath, 1975*; *Randall & Val, 1995*), stomatal movement and development (*Hu et al., 2010*; *Engineer et al., 2014*; *Hu et al., 2015*; *Wang et al., 2016*), biotic and abiotic stress responses (*Slaymaker et al., 2002*; *Restrepo et al., 2005*; *Yu et al., 2007*; *Jung et al., 2008*; *Tianpei et al., 2015*; *Jing et al., 2019*; *Zhou et al., 2020*), and lipid and fatty acid biosynthesis (*Price et al., 1994*; *Hoang & Chapman, 2002a*, *2002b*).

CAs can be classified into several evolutionarily independent families based on their conserved nucleotide sequences, including alpha CAs (α), beta CAs (β), gamma CAs (γ) and delta CAs (δ) (*Hewett-Emmett & Tashian, 1996*; *Liljas & Laurberg, 2000*). Higher plants contain three evolutionarily distinct CA families, and each family is represented by multiple isoforms (*Moroney, Bartlett & Samuelsson, 2001*). A similar number of genes are present in the genomes of plant species as diverse as mosses, monocots, and dicots. For example, *Arabidopsis* contains 19 carbonic anhydrase genes (8 αCA, 6 βCA, 5 γCA), and rice too has a similar number (9 αCA, 3 βCA, 4 γCA) (*DiMario et al., 2017*). However, there are 25 genes coding for CA in soybean due to a past genome duplication (*DiMario et al., 2017*).

Alternative splicing (AS) is an important regulatory mechanism that substantially expands eukaryotic transcriptome and proteome diversity and represents an additional level of cellular regulation (*Naftelberg et al., 2015*). In many species, pre-mRNAs are alternatively spliced during different developmental stages or under stress conditions, allowing organisms to reprogram their regulatory networks (*Mastrangelo et al., 2012*; *Staiger & Brown, 2013*; *Laloum, Martín & Duque, 2018*). AS has been found in some CA family genes. In human, CA IX, one of the 12 enzymatically active carbonic anhydrase isoforms, includes one spliced variant lacking the catalytic domain at the C-terminal, which functionally interferes with the full-length CA IX protein (*Barathova et al., 2008*). AS was also occurred in another human CA gene, which led to produce two different spliced protein forms with both linked to the aggressive behavior of cancer cells (*Haapasalo et al., 2008*). Several isoforms of βCA from *Arabidopsis*, including *AtβCA1*, *AtβCA2*, and *AtβCA4*, are predicted to encode at least two mRNA transcripts *via* AS (*DiMario et al., 2017*). Two spliced variants of *βCA1* have been suggested to be present in *Arabidopsis* (*Oh et al., 2014*; *Rudenko et al., 2017*). Further studies showed that the two spliced transcripts of *AtβCA4* are expressed with different patterns: *AtβCA4.1* is only expressed in leaves, while *AtβCA4.2* is expressed in both roots and leaves (*Aubry et al., 2014*; *DiMario et al., 2016*). AS of CA transcripts was also detected in *Neurachne munroi*, in which four *βCA* transcripts derived from two genes were expressed by AS (*Clayton et al., 2017*). Although AS of CA transcripts is common, the knowledge of AS in CA family genes is still quite limited, especially the identification and molecular characterization of spliced variants of individual isoform.

In this study, we investigated AS in *βCA1* homologs between plant species, focusing on βCA1 spliced variants from *Arabidopsis*. We examined the expression patterns and subcellular localization of individual spliced variants of βCA1. We also analyzed the role of

βCA1 in response to abiotic stress and the protein-protein interaction (PPI) network. This study will contribute to our understanding of βCA1.

## MATERIALS & METHODS

### Plant material and stress treatment

All *Arabidopsis* plants generated in this study are of the Columbia-0 (*Col-0*) ecotype. The T-DNA mutants lines used in this study have been described previously: *βca1* (Salk_106570) and *βca2* (CS303346) (*Huang et al., 2017*). Seeds were surface sterilized and grown on plates with Murashige and Skoog (MS) plus 1% sucrose and 0.8% phyto agar at 22 °C under a long-day (16/8 h light/dark) photoperiod with a photon flux density of 180 μmol photons m-2 s-1. 7-d-old seedlings from the plates were transferred to the soil and grown in the growth chamber. Abiotic stress and phytohormone treatments were subjected to 2-week-old seedlings. For abscisic acid (ABA) 10 μM of ABA was supplied on the solid agar medium with seedling grown vertically. For salt stress, seedlings were vertically placed in plates with 150 mM NaCl for the indicated period of time. For heat treatment, seedlings were transferred to a new plate and placed within a bake oven set at 40 °C for the indicated period of time. For analysis of the seed germination after heat treatment, seeds were sown in the dark for 2 d at 4 °C, and then the seeds were heated at 50 °C for 3 h or 6 h. After this heat treatment, the seeds were grown at 22 °C for 7 days, and then the germination rate and seedling with green cotyledon were measured.

### RNA extraction and reverse transcription-polymerase chain reaction (RT-PCR)

Total RNA was extracted and purified from the 1-week-old *Arabidopsis* seedlings using ReliaPrep™ RNA Miniprep System (Promega, Madison, WI, USA). For cDNA synthesis, the first-strand cDNA was synthesized from 2 μg total RNA with NovoScript®Plus All-in-one 1st Strand cDNA Synthesis SuperMix (Novoprotein, China) according to the instructions. For semi-quantitative RT-PCR analysis primers were designed manually after identification of unique sites in the cDNA of four spliced variants of *βCA1* (*βCA1.1*, *βCA1.2*, *βCA1.3* and *βCA1.4*). Specific primers for amplification of *βCA1.2* and *βCA1.3* were applied, respectively. The primers used were mentioned in Table S1. The *ACTIN2* was used as the endogenous control.

### Identification and analysis of βCA1 in plants

To identify ortholog(s) of βCA1 from *Arabidopsis* in the representative plant genome (including *Oryza sativa*, *Glycine max*, *Populus trichocarpa*, *Brachypodium distachyon*, *Sorghum bicolor* and *Physcomitrella patens*), the amino acid sequence of AtβCA1 was used as the guide sequence to perform a BLASTp search of the database whole genome sequences in the Phytozome database (https://phytozome.jgi.doe.gov). Coding sequences (CDS) and corresponding genomic DNA (gDNA) sequences of the βCA1s from the selected plant species were retrieved from the databases. The GSDS tool (GSDS v2.0, http://gsds.gao-lab.org/) was employed to analyze the exon-intron structures for plant *βCA1* genes.

## Plasmids construction and plant transformation

All DNAs and cDNAs were amplified using Tks Gflex™ DNA Polymerase (Takara, Kusatsu, Japan). The CDS of four *βCA1* spliced variants (*βCA1.1*, *βCA1.2*, *βCA1.3* and *βCA1.4*) were amplified and cloned into pDONOR221 using Gateway™ BP Clonase™ Enzyme Mix (Thermo Scientific, Waltham, MA, USA). Each βCA1 spliced variant was further cloned into the pGWB405 plant expression vector designed for the production of C-terminal GFP-tagged fusion proteins under the control of the 35S Cauliflower Mosaic Virus (35S CaMV) promoter using Gateway™ LR Clonase™ II Enzyme mix (Thermo Scientific, Waltham, MA, USA). To generate the plasmid for the native expression of βCA1, the genomic region including 1.5 kb upstream of ATG plus introns and exons were cloned. For insertion of specific tag including HA or Myc, overlapping PCR was applied by designed primers. These DNA fragments were cloned into pDONOR223 and pGWB4 vectors with the same method used for βCA1 spliced variants cloning. The constructs were confirmed by sequencing. All the plant expression constructs were introduced into the *Agrobacterium tumefaciens* GV3101 strain. Plant transformation was performed with floral dipping method (*Clough & Bent, 1998*). All the primers used for cloning were listed in Table S1.

## Total protein extraction and western blotting

To analyze the GFP-tagged βCA1 spliced variants, seedlings of stable transgenic plants expressing the corresponding GFP fusions were freeze grounded into powder and homogenized in total protein buffer (20 mmo1/L Tris-HCl (pH 7.5), 150 mmol/L NaCl, 2 mmol/L EDTA) with protease inhibitor cocktail in DMSO (Yeasen, China). Lysates were incubated on ice for 20 min and clarified by centrifugation at 18,407 g for 15 min at 4 °C. For immunoblotting, samples were separated on 12% SDS polyacrylamide gel and transferred to PVDF membranes. The membranes were blocked with 5% (g/v) defatted milk in TBST buffer (10 mM Tris-HCl (pH7.4), 150 mM NaCl, 0.05% Tween 20) and probed with using appropriate antibodies including 1:6,000 dilution α-GFP conjugated with HRP (MBL, USA), 1:2,000 α-HA (MBL, Ottawa, IL, USA) and 1:2,000 α-Myc (MBL, Ottawa, IL, USA) overnight at 4 °C. Then the samples were washed with TBST buffer for three times and visualized by using the ECL (Amersham™, USA). Actin protein was used as the internal control.

## Subcellular localization analysis

The protoplasts from the 4-week-old stable transgenic plants leaves with GFP-tagged βCA1 were isolated following the previous used methods (*Yoo, Cho & Sheen, 2007*). The subcellular localization of the GFP fusion proteins from both the protoplast and seedlings was determined by confocal laser scanning microscopy Leica SP8. The excitation wavelength was 488 nm for GFP and the emission window was set at 500–520 nm.

## Co-immunoprecipitation (Co-IP) and MS analysis

The samples of 0.5 g 7-d-old seedlings tissue were freeze grounded to powder and homogenized in 2 ml IP buffer (50 mM Tris-HCl, pH 7.4, 150 mM NaCl, 1 mM MgCl$_2$,

20% glycerol, 0.2% CA-360, 1× protease inhibitor cocktail in DMSO). Lysates were clarified by centrifugation at 14,000 rpm for 15 min at 4 °C and were incubated with Magnetic GFP beads (MBL, Ottawa, IL, USA) for overnight at 4°C in a top to end rotator. After incubation, the beads were washed five times with ice-cold washing buffer (50 mM Tris-HCl, pH 7.4, 150 mM NaCl, 1 mM $MgCl_2$, 20% glycerol and 0.2% CA-360) and then eluted by boiling in reducing SDS sample buffer. Samples were separated by SDS-PAGE. For mass spectrometry analysis, the gel was stained with Coomassie blue after SDS-PAGE separation, and the visible stained protein bands were cut out for in-gel trypsin digestion, followed by tandem liquid chromatograph-mass spectrometry (LC-MS/MS) at Shanghai Applied Protein Technology Co. Ltd.

### Proteins enrichment and protein-protein interaction analysis

Annotation and GO enrichment analysis for proteins identified by MS was performed using the Metascape tools, a free online platform for data analysis (*Zhou et al., 2019*). The protein interaction network analysis was applied in STRING Version 11.0 (minimum required interaction score with high confidence of 0.700) (*Szklarczyk et al., 2019*).

### Gene IDs

Sequence data from this article can be found in The Arabidopsis In-formation Resource (http://www.arabidopsis.org/) under the following gene IDs: βCA1 (At3g01500), βCA1.1 (At3g01500.1), βCA1.2 (At3g01500.2), βCA1.3 (At3g01500.3), βCA1.4 (At3g01500.4).

## RESULTS

### AS analysis in *Arabidopsis* βCA1

A genome-wide analysis of the βCA1 gene family was performed based on the complete genome sequences. Using the Phytozome database and the βCA1 from *Arabidopsis* as the guide sequence for a BLASTp search, we first retrieved available βCA1 sequences from representative sequenced genomes, including three monocots (*Oryza sativa, Brachypodium distachyon, Sorghum bicolor*), two dicots (*Glycine max, Populus trichocarpa*), and *Physcomitrella patens*. Further analysis of genome annotation obtained from the Phytozome database revealed that all *βCA1* genes from selected plant species possessed several transcripts resulting from AS (Fig. S1). At least four spliced variants were found for the *βCA1* gene. These spliced variants seemed to result from pre-mRNA AS, mostly *via* the use of either alternative 5′ or 3′ splice site. By analyzing the latest *Arabidopsis thaliana* genome database, four spliced variants of *βCA1* were found: *βCA1.1, βCA1.2, βCA1.3*, and *βCA1.4* (Fig. 1A). A previous study showed that the full-length βCA1 (same as βCA1.2) contains the chloroplast transit peptide at the N-terminus (*Hu et al., 2015*), whereas βCA1.1 and βCA1.3 coded for truncated proteins, lacking the chloroplast transit peptide at the N-terminus and 11 amino acids at the C-terminus, respectively (Fig. 1B). Moreover, the shortest βCA1.4 was truncated at both ends of the protein.

To investigate these spliced variants of βCA1, the expression pattern of each variant was analyzed. It was not possible to obtain RT-PCR primers suitable for *βCA1.1*, so the corresponding transcripts of *βCA1.1* and *βCA1.2* were analyzed together. The same

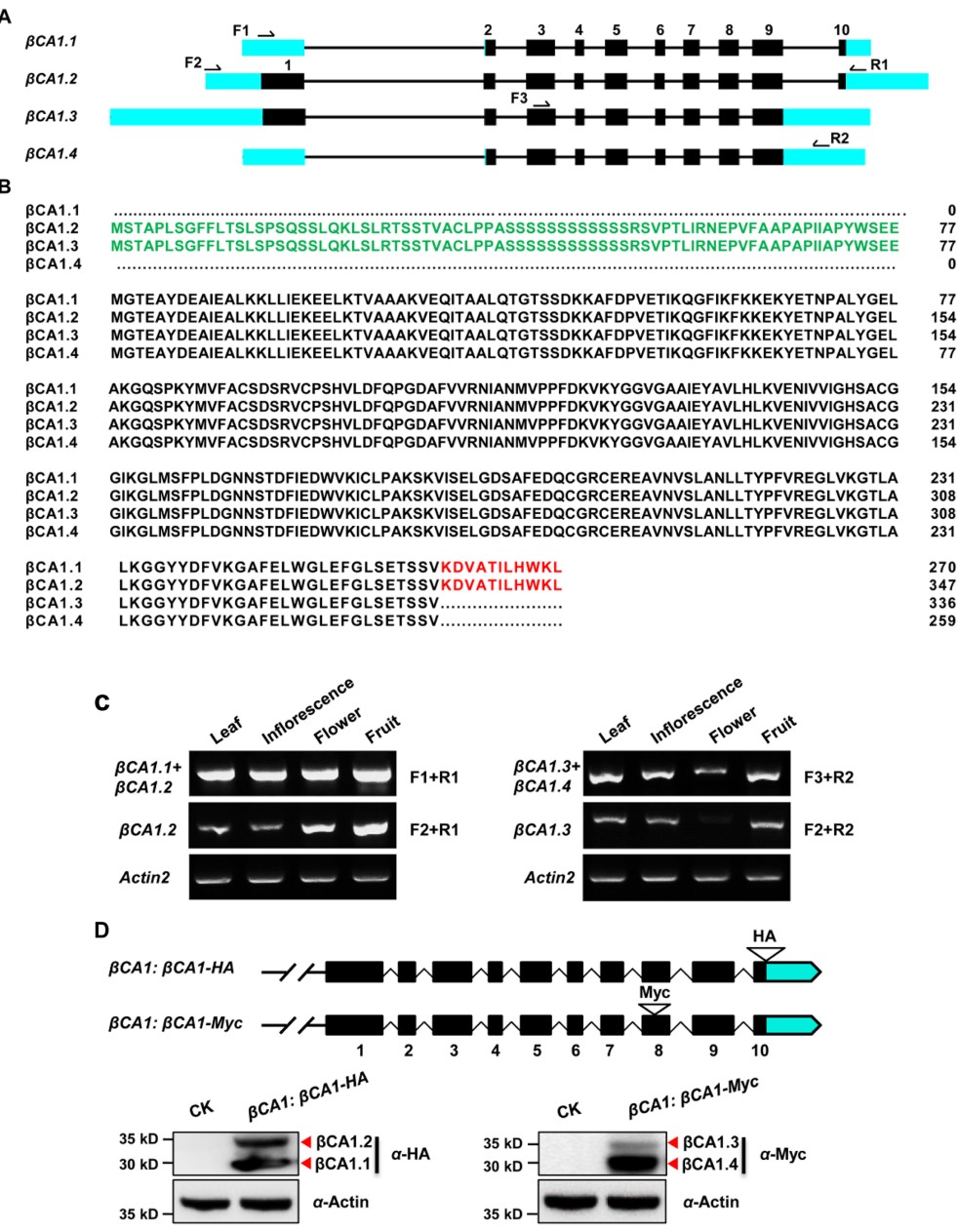

**Figure 1 Sequence structure features of the *Arabidopsis* βCA1.** (A) The exon-intron structures of βCA1 spliced variants were generated by comparing the coding sequences and the corresponding genomic sequences using the GSDS website (http://gsds.gao-lab.org/). The black boxes represent exons, solid lines represent introns, and bright blue boxes represent untranslated regions (UTRs). Primers used for RT-PCR were indicated by arrows. (B) Amino acids alignment analysis for βCA1 spliced variants. The chloroplast transit peptide was in green and the 11 amino acids at the C-terminal were in red. (C) Reverse transcription-polymerase chain reaction (RT-PCR) validation of expression profiling of representative βCA1 spliced variants in different tissues. The RT-PCR products were obtained as: F1+R1 primers for βCA1.1 + βCA1.2, F2+R1 primers for βCA1.2, F1+R2 primers for βCA1.3 + βCA1.4, and F2+R2 primers for βCA1.3. (D) Expression profiling of βCA1 spliced variants. The protein accumulation of βCA1.1 and βCA1.2 was analyzed by western blotting with an anti-HA antibody from the βCA1:βCA1-HA seedlings. And the proteins of βCA1.3 and βCA1.4 were detected with an anti-Myc antibody from the βCA1:βCA1-Myc seedlings. The wild type plants (CK) was used as control.

strategy was applied to *βCA1.3* and *βCA1.4*. For instance, primers F1 and R1 were used for the total transcripts of *βCA1.1* and *βCA1.2*, whereas F2 and R1 were used for the analysis of *βCA1.2* (Fig. 1A). We investigated the transcript expression in four tissues from flowering plants: leaves, inflorescences, flowers, and fresh fruits (Fig. 1C). *βCA1.1* showed the highest global expression levels, with more transcripts from leaves and inflorescences. On the contrary, the expression of *βCA1.2* was higher in flowers and fruits than in leaves and inflorescences. The expression levels of *βCA1.3* were weaker than the expression levels of *βCA1.4*, and *βCA1.3* transcripts were rarely detected in flowers.

To further study the accumulation of each βCA1 spliced variant *in vivo*, two plasmids were generated. The first (*βCA1:βCA1-HA*) had an HA tag (YPYDVPDYA) directly attached to the last exon before the stop codon. The second (*βCA1:βCA1-Myc*) had a Myc tag (QILFRDEFLL) linked to the end of the second-to-last exon, which was designed for expression analysis of four individual spliced variants of βCA1 *in vivo* (Fig. 1D). The spliced variant of βCA1 was detected in the protein extracts from stable transgenic seedlings. The total translation accumulation of βCA1.1 and βCA1.2 were comparable, and protein levels of βCA1.4 were much higher than βCA1.3. However, it is not possible to compare protein levels of all four spliced variants of βCA1 due to the different genetic backgrounds. Taken together, these results suggest that βCA1 is regulated by AS.

### Subcellular localization of spliced variants for βCA1

As the chloroplast targeting peptide was absent in βCA1.1 and βCA1.4, we speculated that the different spliced variants might have different subcellular locations. We then investigated the subcellular localization of each βCA1 spliced variant tagged with GFP in stable transgenic plants. In 1-week-old seedlings, βCA1.1 and βCA1.2 showed a similar distribution with a visible GFP fluorescence signal in leaf chloroplasts localized on stomata (Figs. 2A and B). βCA1.3 was also localized in leaf chloroplasts, however, it seemed to accumulate in the chloroplast envelope and was also observed within the protoplast (Fig. 2B). Surprising, βCA1.4 presented a very different pattern. Strong GFP fluorescence signals in the cytoplasm were observed in transgenic plants with βCA1.4-GFP, whereas no visible signal was detected in chloroplasts either from stomata or leaf protoplasts.

In the root tips, a strong fluorescent signal was detected only for βCA1.4, whereas βCA1.1, βCA1.2, and βCA1.3 were almost undetectable (Fig. 2C). Instead, the latter three variants seemed to accumulate only in small, punctuated structures in the cytoplasm of the root mature zone (Fig. 2D). Subsequent analysis showed that the signals from these small punctuated structures were dynamic (Fig. S2). Immunoblotting analysis confirmed the accumulation of the four spliced variants in the corresponding tissues and organs (Fig. S3).

### Stress influences the transcript abundance of βCA1 spliced variants

The CA1 family is induced under multiple abiotic stresses and phytohormones; therefore, we investigated the effect of some abiotic stresses (salt, heat) and ABA on the expression pattern of *βCA1* spliced variants. The results showed that none of the stresses affected the two βCA1 spliced variant pools (*βCA1.1+βCA1.2*, *βCA1.3+βCA1.4*), whereas salt and

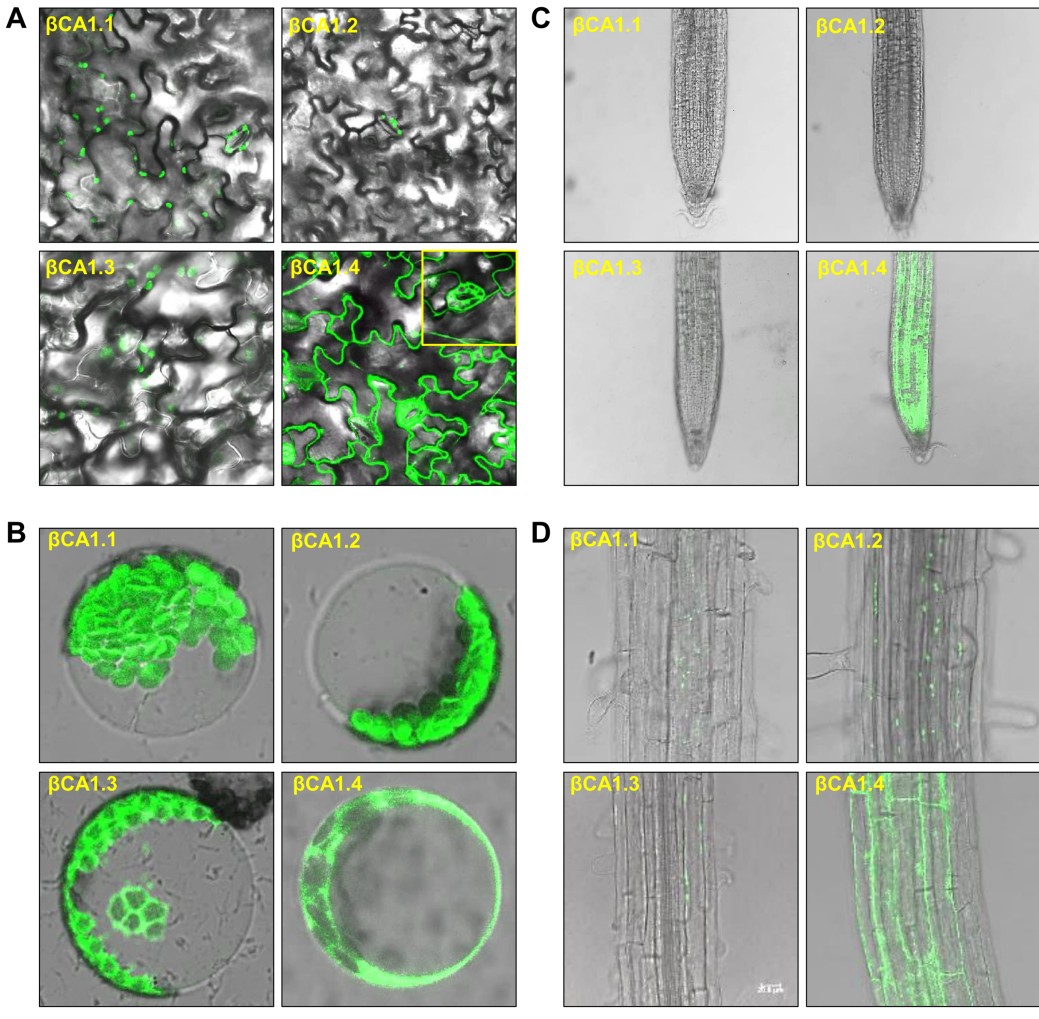

**Figure 2 Subcellular localization of βCA1 spliced variants in transgenic plants.** (A–D) Representative confocal microscopy images were shown in leaves (A), protoplasts (B), root tips (C), and the mature zones of roots (D). The samples of leaves, root tips were from 1-week-old seedlings, and the protoplasts were isolated from the leaves of 4-week-old seedlings.

heat stresses significantly repressed the transcriptional expression of *βCA1.2* and *βCA1.3* (Fig. 3A). This indicated that salt and heat stresses significantly enhanced the induction of *βCA1.1* and *βCA1.4*.

Previous studies had reported that a mature rice CA protein could confer the heat stress tolerance in *E. coli* recombinants and that CA1 proteins from poplar were induced under heat stress (*Tianpei et al., 2015*; *Shi et al., 2017*). Thus, we characterized the response of a T-DNA insertion mutant *βca1* to heat stress (Fig. S4). For the heat stress germination assay, the vernalized Col-0 and *βca1* mutant seeds were pretreated at 50 °C for 3 or 6 h before moving them to normal growth conditions. The results showed that the germination rate of Col-0 was almost unaffected after heat treatment for 3 h, while it decreased to 85% in the *βca1* mutant (Fig. 3B). Under 6 h heat stress treatment, a more significant reduction in the germination rate of *βca1* was observed, whereas 80% of Col-0

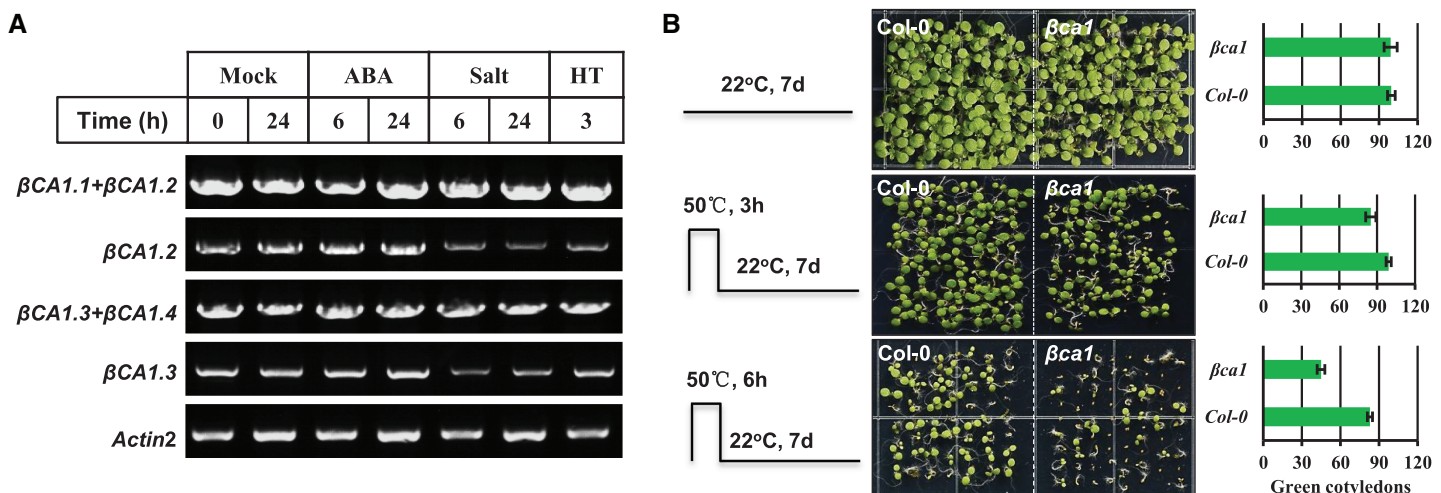

**Figure 3 Functional analysis of *Arabidopsis* βCA1 in response to heat stress.** (A) RT-PCR validation of gene expression patterns of representative *βCA1* spliced variants in the absence (Mock) or presence of stresses. For various stresses treatment, 1-week-old seedlings were subjected to 10 μM ABA, 150 mM NaCl (salt) and 40 °C (HT) on plates, respectively. (B) Representative images of wild-type (Col-0) and *βca1* mutant seedlings without or with heat treatment. Seedlings with green cotyledons were calculated. 

seeds still germinated. This suggests that the *βca1* mutant is heat-sensitive during seed germination.

## Protein-protein inertaction (PPI) network analysis for *Arabidopsis* βCA1

Knowledge of all direct and indirect interactions between proteins will provide new insights into the complex molecular mechanisms inside a cell. To better understand the roles of βCA1 in Arabidopsis, we used co-IP to identify proteins that interact with the most redundant spliced variant βCA1.4. The results showed with high confidence that 109 proteins interact with βCA1.4 directly or indirectly (Table S2).

Among the interacting proteins of βCA1.4, two other βCA family proteins, βCA2 and βCA4, were identified. This interaction was confirmed *in vivo* by bimolecular fluorescence complementation (BiFC) (*Huang et al., 2017*). To gain further insight into their protein functions, the 109 proteins interacting with βCA1.4 were analyzed using Gene Ontology (GO) classifications using Metascape. The analysis revealed that 19 clusters were significantly enriched (*P* value cutoff of 0.01). These proteins played various roles, most of which were related to the functions of βCA1, including photosynthesis and stress response (Fig. 4A). Surprisingly, 29 of these proteins were predicted in response to cadmium ions, which indicated that *Arabidopsis* βCA1 may also function as a cadmium enzyme. In addition, GO analysis showed that 17 proteins, including βCA1, were involved in the response to temperature, which suggests a potential molecular basis for βCA1 in the heat stress response (Fig. 4B).

Additional PPI networks for βCA1.4 associated proteins were constructed using Search Tool for the Retrieval of Interacting Genes/Proteins (STRING 11.0). The full set of 109 proteins produced an interactome map composed of 108 nodes (proteins) and 516 edges, with an average node degree of 9.56 (avg. local clustering coefficient = 0.7) (Fig. 5).

**A**

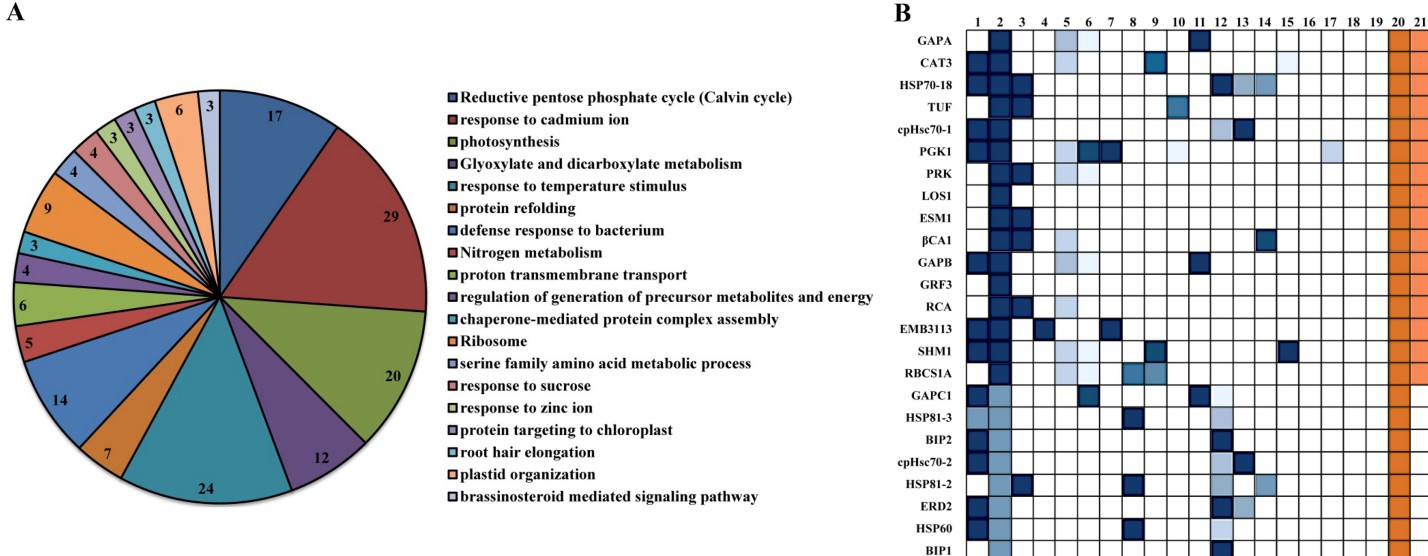

**B**

**Figure 4 GO analysis of *Arabidopsis* βCA1.4 interacting proteins.** (A–B) By using Metascape tool, top 19 clusters with their representative enriched terms (one per cluster) were presented (A). In which, 17 proteins were enriched in response to temperature including βCA1 (B). The Clusters were indicated by numbers, in which 20 and 21 represented "response to temperature stimulus" and "response to cold", respectively.

The various PPI sub-networks are associated with photosystem (green), cellular response to stress (red), and ribosome pathway (cyan), which was consistent with the results from Metascape analysis. Further subcellular localization analysis by STRING showed that most of these proteins are localized in chloroplasts and plastids, where βCA1 proteins are also common. Moreover, several proteins, including βCA1, were predicted to be present in stromules (*Hanson & Hines, 2018*), implying that they may function in morphological maintenance and stromule regulation of non-mesophyll plastids (Fig. S5).

## DISCUSSION

The CA family plays various roles in many biological processes. Both mammalian and plant cells mainly possess two (α and β) or three (α, β, and γ) families with multiple isoforms. In each family, AS within individual isoforms further generates diversity, which significantly contributes to the structural and functional diversification of CA proteins. However, there is still a lack of information regarding AS of CA in plants.

The β family CAs are the most abundant with high expression intensity in plant leaves. Here, βCA1 from the model plant *Arabidopsis* was selected for a comprehensive study. We found that the *βCA1* gene contained at least four spliced variants from all the representative plants, indicating a conserved evolutionary pattern for AS in *βCA* genes (Fig. S1). Interestingly, different expression patterns of spliced variants in *CA* have been observed. In *Arabidopsis*, most of the spliced variants of *βCA2* were detected in leaves using qRT-PCR and the spliced forms of *βCA4* were found to be differentially expressed in tissues (*DiMario et al., 2016*; *Wang et al., 2014*). Alternatively spliced *βCA* transcripts were also detected in the leaves of *N. munroi* (*Clayton et al., 2017*). Here, we found different patterns of transcriptional expression in the *βCA1* spliced variants of *Arabidopsis* (Fig. 1C).

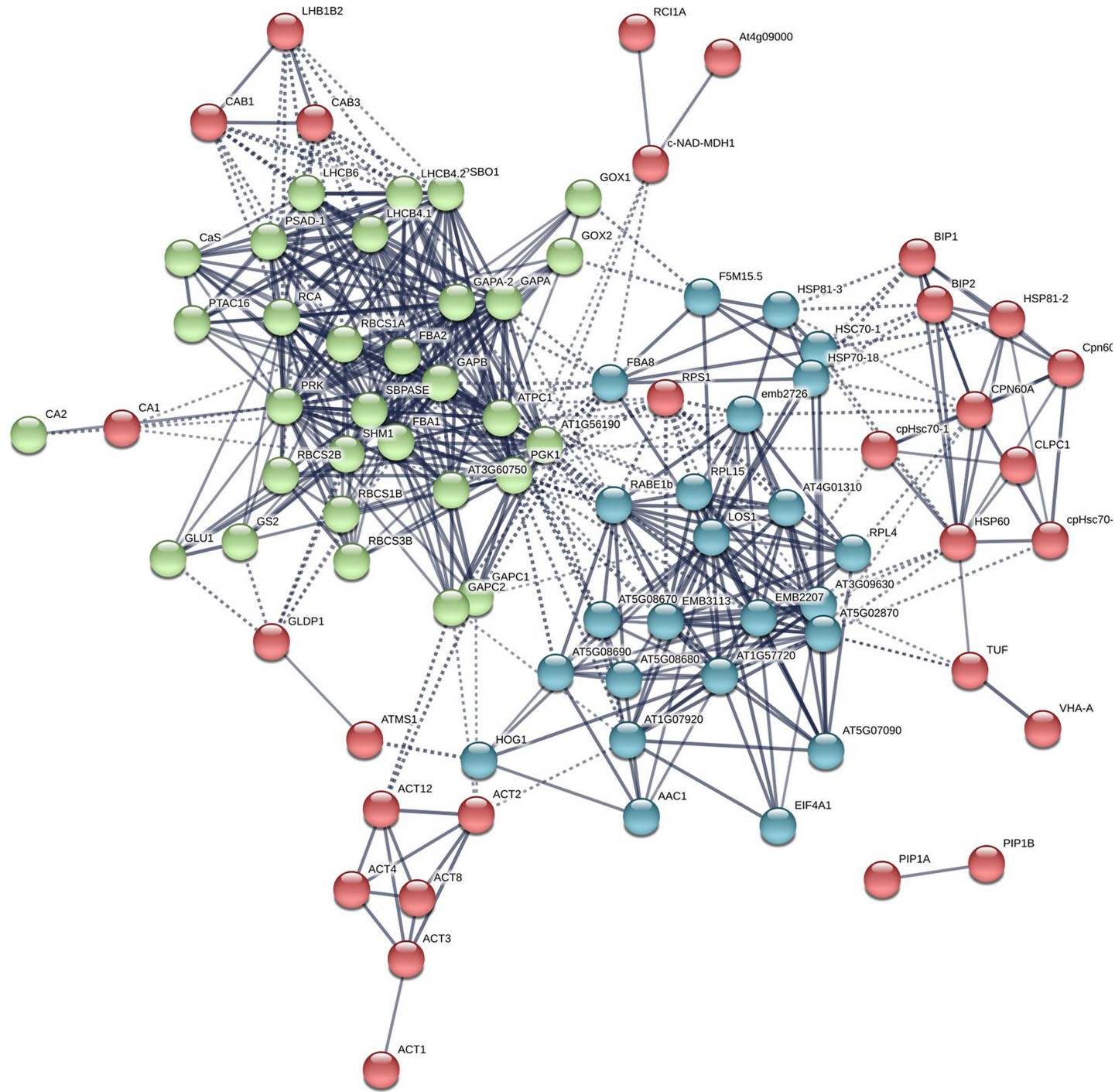

**Figure 5 The PPI of *Arabidopsis* βCA1.4 interacting proteins.** The PPI sub-networks associated with photosystem (green), cellular response to stress (red) and ribosome pathway (cyan) were shown.

*βCA1.1* was found to be the dominant expressed spliced variant. In contrast, *βCA1.3* was weakly expressed and was almost undetectable in flowers. We further investigated whether these spliced variants would undergo accurate translation into proteins. Immunoblotting assays showed the presence of translated proteins for four spliced

variants, indicating they may function *in vivo* (Fig. 1D). The previous study had suggested cleavage sites of the chloroplast targeting sequence in *Arabidopsis* βCA1 (*Hu et al., 2015*), indicating that the mature βCA1.1 and βCA1.2 proteins and the mature βCA1.3 and βCA1.4 proteins are the same size (due to identical peptide content). However, they ran at two different protein sizes on their respective gels (Fig. 1D). We suspected that two bands detected from the *βCA1:βCA1-HA* (or *βCA1:βCA1-Myc*) seedlings might be due to posttranslational modification of βCA1 variants, as βCA1 has been reported to be phosphorylated by the leucine-rich repeat receptor-like kinase EXCESS MICROSPOROCYTES1 (EMS1) (*Huang et al., 2017*). However, the results from immunoblotting analysis were the same with the extracts from *βCA1:βCA1-HA* (or *βCA1:βCA1-Myc*) seedlings in the absence or presence of alkaline phosphatase, suggesting two different protein sizes not result from phosphorylation modification (data not shown). On the other hand, different expression pattern of βCA1 variants were observed. The subcellular localization of them could be same if the mature βCA1.3 and βCA1.4 proteins are the same size. However, our results showed that localization of βCA1.3 is quite different from that of βCA1.4 (Fig. 3). In addition, βCA1.1 was almost undetectable in root, while strong signals of βCA1.1 were observed in leaf. On the contrary, βCA1.2 was weakly expressed in leaf but much strong in root compared with that of βCA1.1 or βCA1.3. Taking together, these results indicated that AS could result in different variants of functional proteins of βCA1 in plant.

The correct inter- and intracellular placement of CAs is essential for efficient physiological function. A previous study showed that the long form of βCA4, βCA4.1, localizes to the plasma membrane while the short form, βCA4.2, is cytosolic (*DiMario et al., 2016*; *Fabre et al., 2007*). This suggests that different spliced variants may show remarkable location diversity, which may enable them to fulfill specific roles. Thus, the subcellular localization of individual spliced variants of βCA1 was investigated. Amino acid sequence alignment analysis showed that βCA1.1 and βCA1.4 were short of the chloroplast transit peptide (Fig. 1B). Consistent with previous report (*Hu et al., 2015*), in our stable transgenic plants βCA1.1 was observed specially localized in chloroplast (Figs. 2A, 2B) and the shortest spliced variant βCA1.4, which is 11 amino acids shorter at the C-terminal in compared to βCA1.1, was mainly targeted in the cytoplasm. This implies the contribution of this short peptide at C-terminus for chloroplast targeting. Eleven amino acids present in the N-terminal region of the β-carbonic anhydrase *NmuCA1a* from *Neurachne* are important for chloroplast targeting and have been suggested to be functional chloroplast transit peptides (*Clayton et al., 2017*). However, the short peptide from *Arabidopsis* (KDVATILHWKL) was different from that in *Neurachne* (ASLGTPAPSSS) and was possibly involved in chloroplast transition. In addition, the short peptide from *Arabidopsis* is only found in CA genes from dicots, while the other one from *Neurachne* is known only from monocots, indicating evolutionary differences between these plants (Fig. S6). This remains to be investigated further. At the organ/tissue levels, only strong expression of βCA1.4 was only observed in mature leaves and root tips (Figs. 2A, 2B). In contrast, localization of the other three spliced variants in the root tips was quite different from that of βCA1.4. As well as being present in chloroplasts,

plasma membranes, and cytoplasm, CA proteins such as βCA6 from *Arabidopsis* and βCA2b from *Neurachne* have also been observed being imported into mitochondria (*Clayton et al., 2017*; *Fabre et al., 2007*; *Jiang et al., 2014*). Moreover, in the red alga *Gracilariopsis chorda*, some CAs showed multiple subcellular locations such as the ER, mitochondria, vacuole, and cytosol (*Razzak et al., 2019*). As the subcellular location of homologous or orthologous genes may be conserved, it would be interesting to further investigate the exact subcellular localization of these βCA1 spliced variants in the root tip. To finding alternatively spliced variants are resulting in different βCA1 proteins that have different subcellular locations is important because: (1) CA proteins may be more redundant within subcellular locations (*i.e.* there may be more CAs in a subcellular location that can compensate for loss of CA activity due to mutation) and (2) this may bolster the potential physiological roles of CA.

The CA gene family has been linked to the stress response of plants. A previous study showed that the transcription level and enzyme activity of the CA family could be induced by various biotic and abiotic stresses, including pathogens, insect herbivores, salinity, and severe temperature (*Yu et al., 2007*; *Wang et al., 2009*; *Collins et al., 2010*; *Kravchik & Bernstein, 2013*; *Chen et al., 2020*). *Arabidopsis* βCA1 has been reported to function together with βCA4 in disease resistance, and was induced under cold stress at the translational level (*Jing et al., 2019*; *Zhou et al., 2020*). These data suggest that the *Arabidopsis* βCA1 plays an important role in the response of plants to environmental stress conditions. However, it is still unknown whether or how these spliced variants of *Arabidopsis* βCA1 respond and contribute to the protection of organisms under different stress conditions. In the present study, *βCA1.1* and *βCA1.4* were induced by salt and heat stresses, whereas *βCA1.2* and *βCA1.3* were significantly repressed (Fig. 3A). This implies that βCA1 is possibly in response to salt and heat stresses. However, no significant difference was observed between wild-type and *βca1* mutants under salt stress (data not shown). On the other hand, we found that loss-of-function *βca1* mutants were much more sensitive to heat stress during seed germination (Fig. 3B). We also analyzed the stress response of *βca2*, the closed homolog of βCA1. However, no obvious phenotype was found in *βca2* under same stress conditions (data not shown). This indicates functional diversity in the CA family. Additionally, the contribution to heat stress tolerance of CA has been found in other plant species, such as rice and *Pyropia haitanensis* (*Yu et al., 2007*; *Tianpei et al., 2015*; *Shi et al., 2017*).

CA associates other proteins by transmitting signals to activate downstream events. Here, we found that the most redundant spliced variant βCA1.4 immunoprecipitated at least 100 proteins in seedlings. Several proteins respond to heat stress, including heat shock proteins, the antioxidant enzyme catalase 3, and peroxidase 45. Heat stress has been found to increase oxidative damage in cells, whereas antioxidant enzymes can lead to protection. This implies that the βCA1 protein could protect cells from damage by reducing peroxides to some extent under high-temperature stress. On the other hand, several studies have shown that coordination between CA and aquaporins could facilitate $CO_2$ transport (*Terashima & Ono, 2002*; *Uehlein et al., 2003*; *Uehlein et al., 2008*; *Yaneff et al., 2014*; *Zhao et al., 2017*). For instance, aquaporin PIP2;1 in *Arabidopsis*

interacts with βCA4 to facilitate $CO_2$ permeability across the plasma membrane (*Wang et al., 2016*). Aquaporin NtAQP1 from tobacco plants has also been identified as a $CO_2$ pore not only in the plasma membrane but also in the inner chloroplast envelope membranes (*Uehlein et al., 2003*). In this study aquaporin PIP1;2 was identified as the iterator of βCA1.4. Comprehensive interaction between CAs and aquaporins may be present in plants and will be interesting to investigate in the future.

## CONCLUSIONS

The βCA gene family has been the most intensely studied in plants. Here, we found that *Arabidopsis* βCA1 was involved in the heat stress response and AS of *βCA1* genes from different plant species. Further investigation showed that four spliced variants of *Arabidopsis* βCA1 were differentially expressed in tissues and in response to stresses. In particular, βCA1.4 showed different localization compared to the other three variants, probably due to loss of a chloroplast transit peptide at the N-terminal and a short peptide at the C-terminal. Interestingly, a short peptide at the C-terminal was specifically appeared in dicots. A more detailed understanding of βCA1 spliced variants will enable us to build a basic working model of their function. Meanwhile, several proteins that interact with βCA1 were identified, which established a basis for further research on the CA family and PPI network in order to understand their physiological roles.

## ACKNOWLEDGEMENTS

We thank Professor Dazhong Zhao (University of Wisconsin-Milwaukee, Wisconsin, USA) for providing the mutants *βca1* (SALK_106570) and *βca2* (CS303346).

### Funding

The project was funded by the Key Laboratory of Molecular Design for Plant Cell Factory of Guangdong Higher Education Institutes (2019KSYS006) and the National Natural Science Foundation of China (31801277). The funders had no role in study design, data collection and analysis, decision to publish, or preparation of the manuscript.

### Grant Disclosures

The following grant information was disclosed by the authors:
Guangdong Higher Education Institutes: 2019KSYS006.
National Natural Science Foundation of China: 31801277.

### Competing Interests

The authors declare that they have no competing interests.

### Author Contributions

- Jinyu Shen performed the experiments, analyzed the data, prepared figures and/or tables, authored or reviewed drafts of the paper, and approved the final draft.
- Zhiyong Li conceived and designed the experiments, performed the experiments, analyzed the data, prepared figures and/or tables, authored or reviewed drafts of the paper, and approved the final draft.
- Yajuan Fu performed the experiments, analyzed the data, prepared figures and/or tables, and approved the final draft.
- Jiansheng Liang conceived and designed the experiments, analyzed the data, authored or reviewed drafts of the paper, and approved the final draft.

## DNA Deposition

The following information was supplied regarding the deposition of DNA sequences:

The sequence data from this article is available in The Arabidopsis Information Resource (http://www.arabidopsis.org/) under the following gene IDs: βCA1 (At3g01500), βCA1.1 (At3g01500.1), βCA1.2 (At3g01500.2), βCA1.3 (At3g01500.3), βCA1.4 (At3g01500.4).

## Data Availability

The raw data for uncropped Gel Blot and RT-PCR is available in the Supplementary File.

## Supplemental Information

Supplemental information for this article can be found online at http://dx.doi.org/10.7717/ peerj.12673#supplemental-information.

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
