# Peer review of "Identification and molecular characterization of the alternative spliced variants of beta carbonic anhydrase 1 (βCA1) from Arabidopsis thaliana"

_PeerJ, doi:10.7717/peerj.12673_

## Round 0.1 · original submission · Minor Revisions

The manuscript needs additional work before publication. The reviewers have provided some suggested editorial changes as well as some comments about interpretation of the data.

Reviewer 1 ·

Basic reporting

This is a very interesting study on carbonic anhydrases. The authors provide an interesting set of data covering the bioinformatics analysis and molecular structure of four βCA1 splicing variants. They provide pooled gene expression data for the four βCA1 splicing variants.
• Data establishes that beta-CA1 has 4 alternative forms, or spliced variants.
• The spliced variants are differentially expressed in Arabidopsis tissues.
• Data shows specific localization of the variants to the chloroplast and cytosol.
1. The introduction addresses the important issues to make the reader understand the paper. One main concern I have throughout the paper is the language and grammatical errors, a few of which I highlight below:
a. The abbreviation for Alternative Splicing is defined twice, in Line 68 and Line 86.
b. Line 77 grammatical errors.
c. Line 101, the 7 beginning sentence should be written in full.
d. Line 126 repetitive word…’sequences’.
e. Lines 129 and 146 correct word …’Analyze’.
f. Line 143 plant transformation….
2. The variants were analyzed in combination as (β-CA1.1 and β-CA1.2) and for β-CA1.3 and β-CA1.4. This implies the transcript is in pools as shown in Fig 1C. However the reporting in Lines 208-213 separates βCA1.1 and βCA1.4 as if they are individual expression of individual variants. I think the reporting needs to be consistent for the pools.

Experimental design

The methods are described clearly.
1. One issue I raise is specificity on how the plants were exposed to heat stress. It is not clear if one wanted to repeat this treatment. I refer to Line 105-107: It is not clear whether the seeds or the plants were subjected to heat.
2. On the same matter, there is inconsistency on the temperature used for heat treatment. Line 106 in text mentions 50oC, while Fig 3 caption indicates 40oC, the Image Fig 3 shows 50oC. Please clarify.
Line 179 – 183: While the title refers to accession numbers, the Gene ID or Loci is what is provided not the accession numbers. Either remove the title, or change it to gene ID, or provide actual accession numbers.
On subcellular localization, Line 157 mentions protoplasts from Four-week old seedlings, but Figure 2 is saying one week old seedlings. Please address the inconsistency.

Validity of the findings

No comment

Additional comments

The manuscript needs language and grammatical editing. I recommend a proficient English writer or professional editing service.
a. Additional examples of these errors:
b. Sentence in Lines 76 – 78 not grammatically correct.
c. Sentence Line 149- 151
d. Line 251: The importance of …..
e. Lines 285 -287, sentence not clear.
f. Line 284 …’frechuntely’

RT-PCR gene expression data is great and shows expression differences. Quantitative (q-PCR) would have been more robust and could quantify expression of each individual variant.
3. There is a Table in Fabre et al., (2007) showing that Arabidopsis βCA1 has 9 cDNAs. Here you report four splicing variants. So the other cDNAs do not code functional proteins? It might be important to bring this to attention in your discussion.

Reviewer 2 ·

Basic reporting

Although the manuscript flows well, a significant amount of the text has English grammar and sentence structure mistakes. I recommend the authors use an English editorial service to help refine the text. Here are some (not all) examples where the text can be edited:
1. Please be careful of using italics (βCA1 gene) or not (βCA1 protein).
2. Line 32 – Please remove, “study of…” so the line reads, “…we focused on the characterization of…”
3. Line 34 – “variant” should be “variants”
4. Lines 34-36 – This sentence should be reworded to, “The results showed that the spliced variants of βCA1 possessed different subcellular and tissue distributions and responded differently to environmental stimuli.”
5. Line 52 – Please remove “the” so the line reads, “…gas and ion exchange…”
6. Line 55 – Please add, “and” so the line reads, “…and lipid and fatty…”
7. Line 58 – Please change to, “…based on their conserved nucleotide sequences,”
8. Line 62 – Please remove “And” so the sentence starts as, “A similar number of genes…”
9. Lines 65-67 – Please re-word as, “However, there are over 25 genes coding for CA in soybean due to a past genome duplication (DiMario et al., 2017).”
10. Line 89 – Please remove “firstly”
11. Line 101 – Please modify “μmol•m-2•s-1” to “μmol photons m-2 s-1” with the appropriate superscripts.
12. Line 107 – Please remove “kept”
13. Lines 126-127 – Please reword as, “Coding sequences (CDS) and corresponding genomic DNA (gDNA) sequences of the…”
14. Line 199 – Please change “N-terminal” to “N-terminus”
15. Line 200-201 – Please reword as “…lacking the chloroplast transit peptide at the N-terminus and 11 amino acids at the C-terminus, respectively.”
16. Lines 201-202 – Please reword as, “Moreover, the shortest βCA1.4 variant was truncated at both ends of the protein.”
17. Lines 203-204 – Please reword as, “…the expression pattern of each variant was analyzed.”
18. Lines 210-213 – Please reword as, “On the contrary, expression of βCA1.2 was higher in flower and fruit tissues than in leaves and inflorescences. The expression levels of βCA1.3 were weaker than the expression levels of βCA1.4, and βCA1.3 transcripts were rarely detected in flowers.”
19. Line 216 – Please change “code” to “codon”
20. Line 221 – Please change “it’s” to “it is”
21. Line 226 – Please change “localization” to “locations”
22. Line 228 – Please change to, “In one-week-old seedings…”
23. Line 235 – Please change to, “In the root tips, a strong fluorescent signal was detected only for βCA1.4, whereas…”
24. Line 242 – Please reword as, “The CA1 family is…”
25. Line 244 – Please remove “hormone”
26. Line 245 – Please change “stress was…” to “stresses were…”
27. Line 284 - Please change "frechuntely" to "frequently".
28. Line 291 – Please add “the” to the beginning of the sentence. Also, please remove “in” so the next sentence reads, “Both mammalian and plant cells…”
29. The wording/English of the discussion section can use some refining as well.


The introduction provides a nice background on carbonic anhydrases (CA) and pertinent CA references are sighted for the most part. There are some issues:
1. Should the reference section be organized in alphabetical order?
2. Lines 242-243 - DiMario et al., (2017) is a review article. Should the original research articles be referenced here?
3. Lines 378-379 - The interaction between PIPs and βCA4 in Arabidopsis thaliana has already been confirmed (Wang et al., 2016. The Plant Cell. 28: 568-582).


The article is structured well. However, the figures could be of better quality (higher resolution.) Also, the authors introduce new data in the discussion section (Fig. S5; line 334). It seems that introducing new data in the discussion section is not usually allowed, but I will leave this up to the authors to decide to move this to the results section or not.

Experimental design

The research is a good fit with the Aims and Scope of the journal.


The authors do a good job of identifying a knowledge gap in the plant CA field and their research does address this gap.


There are some issues with the experimental design and how the methods are written:
1. Lines 98-99 - The βCA2 mutant line does not seem to be used in this study. Additionally, Huang et al. (2017) used a βca1βca2βca4 triple mutant and did not characterize the βca1 T-DNA single mutant used in this study.
2. Additionally, with the known problems of secondary mutations with T-DNA mutants, backcrossing or at least complementation lines are necessary controls for growth experiments. Did the authors perform backcrosses on the βca1 T-DNA line? Is it possible to generate complementation lines by inserting a functional AtβCA1 gene back into the βca1 T-DNA line?
3. Line 150 - If reporting in RPM, the centrifuge and rotor information must be provided, too. Or, reporting the centrifuge step in g is an option.
4. Figure 1C - For expression semi-quantification using RT-PCR, it is important to standardize the primer sets where the primer pairs amplify the same size amplicons and are located in similar regions of genes (as 3' ends of transcripts are more easily reverse transcribed versus 5' ends in some cases.) The different locations of the primer sets along the βCA1 gene could not be helped as the authors had no choice there, but it is possible to standardize the size of the different PCR products. I just wonder if amplicon size had an impact on the results of expression strength.
5. For Figure S3 - I may have misunderstood this experimental setup. I do not understand how you can differentiate between the different βCA1 protein variants when using 1.5 kb promoter regions upstream of the different transcription start sites of the variants. These different 1.5 kb promoter regions have significant overlap with one another at the 5' regions, particularly for βCA1.1, βCA1.2, and βCA1.4. How can you be certain that you are not picking up the correct βCA1 variant?
6. Figure 3B - A minor question. Are the wild-type (WT) and βca1 mutant seedlings growing on the same plate, or are they growing on their own independent plates? In the past, have groups tried to grow different genetic lines on the same large agar plates to reduce variability?


Overall, the methods section is written well and descriptively.

Validity of the findings

The GFP and βCA1 protein interaction datasets are very interesting and the authors overall provide an interesting story with their manuscript.

There are some issues with other datasets/conclusions in the manuscript:
1. I am speculating here, but I question the validity of Figure 1D in relation to Figure 1A - According to the alignment in Figure 1A, βCA1.1 and βCA1.2 should have the same molecular weights (and isoelectric points since they are the same sequence) once the chloroplast transit peptide (cTP) has been cleaved from βCA1.2, yet there are two distinct band sizes for βCA1.1 and βCA1.2. The same can be said for βCA1.3 and βCA1.4 as these two proteins should also run on top of each other in the protein gel. I also question the identification of the chloroplast transit peptide in Figure 1A. Running the βCA1 peptide sequence through ChloroP - the chloroplast transit peptide prediction program (http://www.cbs.dtu.dk/services/ChloroP/), the predicted cleavage point is around ALA104 (which has the highest CS-score - most likely residue to be the cleavage point of the cTP). An alanine as the cleavage residue makes more sense than the GLU-MET region indicated as the cleavage point for the cTP. If ALA104 is the true cleavage site of the cTP, then the mature βCA1.2 and βCA1.3 proteins with cleaved cTPs should actually be smaller than than βCA1.1 and βCA1.4.
2. As mentioned above, there are known issues with Arabidopsis T-DNA mutant lines. If the βca1 T-DNA line has not been backcrossed (which was not stated by the authors), then a couple independent complementation lines with a functional βCA1 gene inserted back into the βca1 line is a necessary control for the heat stress growth experiments.
3. Again, the GFP localization data are very nice and very interesting. However, with lines 232-233 and lines 324-325 - I do not think you can say βCA1.4 is localized to both the cytosol and plasma membrane. I agree βCA1.4 is likely in the cytosol, but you will need further experimental proof to say that βCA1.4 is also in the plasma membrane. Due to the central vacuole pushing organelles and the cytosol outward toward the plasma membrane, a cytosol fluorescent signal can look like a plasma membrane signal at times.
4. Lines 325-326 - Are there any references for C-terminal regions being cTPs? I thought cTPs were strictly at the N-termini of proteins?
5. In the Discussion section, it would be nice to point out how important it was to find that alternatively spliced variants are resulting in different βCA1 proteins that have different subcellular locations. This is important because: 1) CA proteins may be more redundant within subcellular locations (i.e. there may be more CAs in a subcellular location that can compensate for loss of CA activity due to mutation) and 2) this may bolster the potential physiological roles of CA.

Additional comments

Again, the GFP localization and βCA1 protein interaction datasets are very nice and interesting.

Reviewer 3 ·

Basic reporting

Manuscript by Shen et al. “Identification and molecular characterization of the alternative spliced variants of beta carbonic anhydrase 1 (βCA1) from Arabidopsis thaliana” describes the study of an important problem of the functioning of plant carbonic anhydrases, whose functions in higher plants are still largely unknown. The authors obtained valuable data on the characteristics of the most abundant in plant leaves carbonic anhydrase, βCA1. The presence of enzyme splicing, and in particular of carbonic anhydrases, reveals new features of the metabolism of living organisms. The presented study of this question for an enzyme such as βCA, whose functions remain mysterious, makes a great contribution to our understanding of these functions.
The structure of the manuscript is standard and contains all the necessary parts. Further analysis of the text will be given separately by sections. However, first about the language. English is grammatically correct, but very difficult to read and understand, the authors need to improve it for clarity. Some obvious errors will be noted further.
Abstract fully reflects the content of the work.
Introduction contains the data necessary to justify the undertaken work, and the references reflect the problems that arise. However, very little space is given directly to the problem of βCA1 functioning in plants. Surprisingly, the authors did not mention a work published as early as 2107 (Rudenko et al. Biochemistry (Moscow), 2017, Vol. 82, No. 9, pp. 1025 1035). In this work, the dependence of the spliced variables of βCA1 on light intensity and photoperiod was investigated, and the latter is important for understanding the results of this manuscript. In that work the expressions levels of two spliced variants of βCA1 were studied, and it was sown that they were different at short (8 h day/16 h night) and long (16 h day/8 h night) photoperiods, the last was used in authors’ study (line 102 in the manuscript).
Materials and methods are described in sufficient detail to be able to reproduce the experiments of the authors. However, there are a number of observations.
Line 119. Tablet S1?? Check it.
Line 124. Replace word ‘seed’ to more appropriate one. The same in the line 189 in Results.
Lines 151-155. The description of immunoblotting is required to be more in detail! Especially about the reagents used.

Experimental design

Materials and methods are described in sufficient detail to be able to reproduce the experiments of the authors. However, there are a number of observations.
Line 119. Tablet S1?? Check it.
Line 124. Replace word ‘seed’ to more appropriate one. The same in the line 189 in Results.
Lines 151-155. The description of immunoblotting is required to be more in detail! Especially about the reagents used.

Validity of the findings

Results. The description of the results obtained is enough clear, but it is desirable that the most important results are more clearly indicated.
Lines 204-206 repeat lines 116-118 in the Methods. The approach would be preferably to present in more detail in the Results.
Line 209. It is necessary to describe how the material from the "inflorescence" was prepared.
Line 220. The word" while" distorts the meaning of the sentence.
Line 245. ‘βCA1.2+βCA1.3’ should be replaced with ‘βCA1.3+βCA1.4’!!
Since co-immunoprecipitation can be non-specific, the data described in the relevant part seem questionable. Such concern confirms by the fact that co-immunoprecipitation for βCA1. 4 was detected with photosystem-bound proteins (line 281), although “no visible signal can be detected in chloroplasts from either stomata or leaf protoplasts” (line 234).
Line 284. "frechuntely" or "frequently"?
Discussion.
This part of manuscript is rather adequately reflected the results obtained. However, some discussion about physiological functions of spliced variants of βCA1 in the different locations is necessary, as it was recommended also for Introduction.
It may be noted such important conclusions as ‘that the individual spliced variant of βCA1 may function in vivo’ (line 309). However, in some places the clarification of authors’ statements is required.
Lines 317 - 323. The authors ' reasons for explaining the difference between their results and the data by Hu et al. (2015) seem rather light and contradictory. A more detailed discussion is needed.
Lines 354 – 355. “In the present study, βCA1.1 and βCA1.4 were found induced by salt and heat 355 stresses, whereas βCA1.2 and βCA1.3 were significantly repressed.’ However, this conclusion was made in Results only as assumption!!
Line 359. ‘βca2’ in italic. Possibly, this is incorrect.
Lines 377 - 378. βCA4.1 should be instead of βCA1.4.
The figures and tables in the main text are suitable.
Very serious. The legends to Tables and figures are absent for supplementary data.

---

## Round 0.2 · Minor Revisions

There a still some minor revisions that should be addressed - please see the comments from reviewer 2.

Please be sure to address the point regarding the ßCA1.3 and ßCA1.4.

Reviewer 2 ·

Basic reporting

Overall, the paper flows well. There are only a few minor typos:
1. Line 28 – in the abstract, “oxide” should be “dioxide”.
2. Line 48 – There should be a space between “tissues” and “across”.
3. Line 98 – in the methods, “soli” should be “soil”.
4. Line 192 – in the results, “an” should be deleted so the line reads, “…alternative 5’ or 3’ splice site.”
5. Line 214 – in the results, “I,” should be deleted.
6. Line 244 – in the results, there should be a space between “the” and “heat”.
7. Line 283 – in the discussion, “various” is used twice in this sentence. Maybe substitute one of the “various” words with “many”?
8. Line 288 – in the discussion, should “redundant” be replaced with “abundant”?

The introduction provides a nice background on carbonic anhydrases (CA) and the pertinent CA references are cited.

The article is structured well. Some figures may need to be of better quality; however, this may be from my PDF viewer. I leave this up to the discretion of the authors and editor.

Experimental design

The research is a good fit with the Aims and Scope of the journal.

The authors do a good job of identifying a knowledge gap in the plant CA field and their research does address this gap.

Overall, the methods section is descriptive and well-written.

Validity of the findings

The GFP and βCA1 protein interaction datasets are very interesting.

I am still concerned that the protein alignments of Figure 1B indicate that the mature βCA1.1 and βCA1.2 proteins and the mature βCA1.3 and βCA1.4 proteins are the same size (and have the same isoelectric points due to identical peptide content), yet they run at two different protein sizes on their respective gels. However, this does not take away from the main points that are being made by this study.

Additional comments

The authors did a fairly good job of responding to the manuscript’s critiques and made the corresponding changes that were needed.

Again, the GFP localization and βCA1 protein interaction datasets are very nice and interesting.

---

## Round 0.3 · Minor Revisions

Thank you for making the suggested changes to the manuscript. I would suggest adding some of the discussion regarding the different sizes of the protein variants in the discussion.

Note that "investigated" (line 303) is misspelled.

---

## Round 0.4 · Minor Revisions

Thank you for making the requested edits to the manuscript. The manuscript is almost ready to move forward. The last few additional suggestions that were missed in prior iterations:

For the GO: terms provided in Figure 4, their numerical code counterparts (GO:12345) should also be provided, either as part of the figure or in a supplemental table.

LINE NO: / BEFORE / AFTER / [COMMENTS]
LINE 103: / oven set as 40 / oven set at 40 / [.]
LINE 103: / analysis the seed / analysis of the seed / [.]
LINE 124: / to analyze of the exon-intron / to analyze the exon-intron / [.]
LINE 388: / . / . / [ Really the most intensely studied, or just for the bCAs?

---

## Round 0.5 · accepted · Accept

Thanks for making the changes.